# The Impact of Anorexia Nervosa and the Basis for Non-Pharmacological Interventions

**DOI:** 10.3390/nu15112594

**Published:** 2023-06-01

**Authors:** Vicente Javier Clemente-Suárez, Maria Isabel Ramírez-Goerke, Laura Redondo-Flórez, Ana Isabel Beltrán-Velasco, Alexandra Martín-Rodríguez, Domingo Jesús Ramos-Campo, Eduardo Navarro-Jiménez, Rodrigo Yáñez-Sepúlveda, José Francisco Tornero-Aguilera

**Affiliations:** 1Faculty of Sports Sciences, Universidad Europea de Madrid, Tajo Street, s/n, 28670 Madrid, Spain; vctxente@yahoo.es (V.J.C.-S.); mariaisabel.ramirez@universidadeuropea.es (M.I.R.-G.); josefrancisco.tornero@universidadeuropea.es (J.F.T.-A.); 2Department of Health Sciences, Faculty of Biomedical and Health Sciences, Universidad Europea de Madrid, C/Tajo s/n, Villaviciosa de Odón, 28670 Madrid, Spain; lauraredondo_1@hotmail.com; 3Psychology Department, Facultad de Ciencias de la Vida y la Naturaleza, Universidad Antonio de Nebrija, 28240 Madrid, Spain; abeltranv@nebrija.es; 4LFE Research Group, Department of Health and Human Performance, Faculty of Physical Activity and Sport Science-INEF, Universidad Politécnica de Madrid, 28040 Madrid, Spain; 5Facultad de Ciencias de la Salud, Universidad Simón Bolívar, Barranquilla 080002, Colombia; eduardo.navarro@unisimon.edu.co; 6Faculty of Education and Social Sciences, Universidad Andres Bello, Viña del Mar 2520000, Chile; rodrigo.yanez.s@unab.cl

**Keywords:** anorexia nervosa, interventions, non-pharmacological, individual profile, therapy, eating disorder, microbiota, physical exercise, nutrition, psychology

## Abstract

Anorexia nervosa is a psychiatric disorder with an unknown etiology that is characterized by an individual’s preoccupation with their weight and body structure while denying the severity of their low body weight. Due to the fact that anorexia nervosa is multifaceted and may indicate the coexistence of genetic, social, hormonal, and psychiatric disorders, a description of non-pharmacological interventions can be used to ameliorate or reduce the symptoms of this condition. Consequently, the purpose of the present narrative review is to describe the profile’s context in the anorexic person as well as the support they would require from their family and environment. In addition, it is aimed at examining preventative and non-pharmacological interventions, such as nutritional interventions, physical activity interventions, psychological interventions, psychosocial interventions, and physical therapy interventions. To reach the narrative review aims, a critical review was conducted utilizing both primary sources, such as scientific publications, and secondary sources, such as bibliographic indexes, web pages, and databases. Nutritional interventions include nutritional education and an individualized treatment for each patient, physical activity interventions include allowing patients to perform controlled physical activity, psychological interventions include family therapy and evaluation of the existence of other psychological disorders, psychosocial interventions include management of the relationship between the patient and social media and physical therapy interventions include relaxation massages and exercises to relieve pain. All these non-pharmacological interventions need to be individualized based on each patient’s needs.

## 1. Introduction

Eating disorders are diseases whose main characteristics are distorted eating behavior and an extreme concern for self-image and body weight. The two main eating disorders are anorexia nervosa (AN) and bulimia nervosa (BN), referring in this review exclusively to AN. According to the diagnostic and statistical manual for mental disorders (5th edition) (DSM-5), individuals with AN are disturbed by their weight and body shape, without recognizing the graveness of their low weight. The prevalence of AN is 0.4% among women and 0.1% among men at any point in time, worldwide [1]. However, recent reviews suggest that the lifetime prevalence rates of AN might be up to 4% among females and 0.3% among males. Regarding BM, up to 3% of females and more than 1% of males suffer from this disorder during their lifetime. According to epidemiological studies, the trend is for a significant percentage increase in AN, this being greater in women than in men [2].

New fashion trends and new standards in physical aspects and eating patterns are indicated as possible triggers for the increase in the frequency of AN. Yet, as with many psychiatric illnesses, the etiology of AN is not yet known with certainty, but is thought to be multifactorial and may include components of genetic, social, metabolic, personality, hormonal, sexual, the way of expressing emotions, learning, abuse history, mistreatment, perfectionism, and the coexistence of other psychiatric disorders (such as a depressive component, obsessive–compulsive, anxiety disorders and loss of impulse control) [3]. The initial consequence of AN is decreased caloric and nutrient intake. With this deficit and malnutrition, disorders appear in different organ systems, including cardiovascular, gastrointestinal, endocrine, neurological and skeletal [4]. These organic dysfunctions in the different systems are accompanied by dysfunctions in the mental health of patients with AN such as mood and anxiety disorders. If AN is considered as a pathology of a psychiatric nature, it is one with the highest mortality rate compared to other mental disorders [4], due to increased suicidal thoughts (up to 10% more). Indeed, one in five deaths associated with AN has been due to suicide [5].

A greater number of resources are needed, including more healthcare professionals to deal with the increased patient-related workloads. In the case of AN, the treatments are highly expensive given the frequency of relapses and the high disease burden [6]. Given the prevalence of the disease and the epidemiological curve, special focus should be made since it can increase the global healthcare costs. Furthermore, from an economic point of view, it is not only the associated health costs, but also the decreased work capacity of subjects with AN, which affect the world economy indirectly, due to a decrease in workplace productivity produced by the physical and mental health implications of AN [7].

Interventions can be of two types: pharmacological treatments and non-pharmacological interventions. The interventions that concern this review are of a non-pharmacological nature, and among most traditional interventions we can find [6]:Family therapy (changing and solving family problems to cure AN)Family based treatment (parents’ involvement in adolescents’ food consumption)Joint family therapy (collaborative work among adolescents, the entire family and a therapist with monitoring of the family’s emotional issues)Behavioral family system therapy (three-step behavioral weight gain program with family involvement)Cognitive-behavioral therapy (modifications of irrational beliefs and problematic eating behavior)Specialist supportive clinical management (Education with supportive therapy)

The aforementioned refers to psychological treatments, however, like any pathology, prevention and treatment from a holistic point of view is essential. Especially since current treatment efficacy remains limited; around 40% of AN patients after 10 years of medical care still show prolonged symptoms and disabilities [8]. This highlights the importance of developing new alternative therapies against AN. Therefore, the inclusion of other areas such as nutrition, physical exercise, and psychosocial aspects among others, are essential and will be discussed throughout this review. Therefore, the present narrative review focuses on the analysis from a holistic perspective of the multifactorial etiology of AN, addressing the different medical–scientific disciplines considered non-pharmacological interventions, to give a broader view of the pathology of AN, generating holistic practical applications that can improve the patient intervention processes.

## 2. Materials and Methods

To reach the narrative review’s aims, a critical review was conducted utilizing both primary sources, such as scientific publications, and secondary sources, such as bibliographic indexes, web pages, and databases. This narrative review focuses on non-pharmacological interventions for the AN patient. The main research topic was combined as follows:

AN or malnutrition or nutritional disorder and psychological profile or microbiome or oral health or physical activity or nutritional intervention or psychological intervention or physical therapy or psychosocial intervention. The search was limited to English manuscripts, excluding grey literature, with a publication range between 2012 and 2023, except for the classic literature. In addition, retrieved articles, practice guidelines, editorials, and letters were searched for additional references. To cover the multifactorial nature of mental health, several databases were used as MedLine, Cochrane, Embase, Psych-INFO and CinAhl. Studies were included if they addressed any topic related to nutrition, physical activity, oral health, mental health, inflammation, gut microbiome, or non-pharmacological interventions. Additionally included were described nutritional interventions that were published in a nutritional/psychiatry/psychology journal. Exclusion criteria were: (i) research outside the time period analyzed, (ii) presented topics out of the review scope, (iii) unpublished studies, books, conference proceedings, abstracts, and PhD dissertations. Information extraction was performed by the authors of this manuscript, who divided the information according to their area of expertise. This divided the text into different fragments that form the narrative line of the present narrative review.

## 3. The Impact of Anorexia Nervosa on an Individual’s Life

### 3.1. Psychological Profile

The individual profile has been highlighted as one of the most important factors which may lead to anorexia development and severity [9]. Thus, understanding the profile that characterizes anorexia patients may increase the likelihood of providing more effective treatments and preventive strategies [10]. Characteristics related to an individual’s profile are perfectionism, obsessive–compulsive disorder, self-esteem, cognitive rigidity, and neuroticism.

Previous literature describes how perfectionism, obsessive–compulsiveness and dysphoric mood could be considered as characteristics strongly related to anorexia [11]. Furthermore, it was described several years ago how anorexia patients could present anxiety persistence, perfectionism, and obsessional behaviors, including symmetry, exactness, and order [12,13,14]. Nowadays, researchers point out how female anorexia patients self-report significantly lower cognitive flexibility and significantly higher clinical perfectionism [15]. In this study, participants self-reported in a questionnaire which evaluated different items, including cognitive, behavioral and affective components of setting aims and struggling to achieve them, as well as the consequences on their self-evaluation when these principles were accomplished or not accomplished [16]. Thus, according to these findings, perfectionists were more prone to anorexia, probably due to the fact that they needed to comply with patterns inferred as role models to be followed by society or their own thoughts.

Obsessive–compulsive disorders are another important aspect related to anorexia, since both have been bidirectionally related in several studies [11,17]. Along these lines, it was described how 10 to 60% of obsessive–compulsive disorder patients suffered from anorexia as well as close to 10 to 40% of anorexia patients were primary obsessive–compulsive disorder diagnosed [17,18,19]. Self-esteem is another key factor which has been associated with anorexia. Thus, it could be defined as a positive or negative attitude concerning oneself [20]. Lower levels of self-esteem have been largely described as a risk factor of anorexia, as well as being pointed out as predisposing and precipitating factors, compromising anorexia prognosis, including treatment and recovery [21,22,23]. Additionally, a strong association was suggested by research conducted in the last two decades, showing how anorexia patients presented low self-esteem values compared to non-eating-disorder patients [24,25,26,27,28].

Neuroticism has also been pointed out as a risk factor which could compromise anorexia status. Neuroticism is, along with extraversion, agreeableness, conscientiousness, and openness to experience, one of the five factor model personality traits, and it could be defined as the opposite term to emotional stability [29,30]. Thus, neuroticism involves a higher sensitivity and a strong tendency to experience negative feelings, as well as a lower resilience against stress [29]. Hence, it has been largely described among recent and prior literature how neuroticism plays a crucial role in body image dissatisfaction and eating related disorders [31,32,33].

Finally, cognitive rigidity has also been proposed by previous researchers for its ability to play an important role in anorexia [34]. Cognitive rigidity could be defined as the tendency to pay attention to one’s own thoughts, beliefs or behaviors, sometimes excluding opinions, thoughts, beliefs or ideas of others [35]. Thus, recent literature described how anorexia patients showed deficits in cognitive flexibility in verbal and nonverbal domains [36]. Moreover, previous authors highlighted how this cognitive rigidity also could be presented by first degree relatives, a fact which may reinforce the obsessive personality in anorexia patients [37,38]. Previous studies related cognitive flexibility to obsessive–compulsive disorder [39] as well as to obsessive–compulsive personality disorder [40,41], conditions which may favor the reinforcement of the view that one’s own ideas are the right ones, leaving aside the opinions of the other people with whom the anorexia patient interacts. Additionally, recent research highlighted an association between perfectionism and cognitive inflexibility in anorexia patients [42]. Thus, according to these findings, cognitive inflexibility may be considered as a potential triggering factor in anorexia, where a patient’s own thoughts predispose them not to consider the opinion of others. Moreover, it could also enhance the idea of perfectionism previously mentioned, where patients need to fulfil role models in order to increase their self-esteem, meeting their own expectations. Furthermore, as we mentioned above, it is important to consider how previous literature correlated anorexia, neuroticism and obsessive—compulsive disorder, a fact that may be explained through biological mechanism shared in the development of the three pathologies. We consider these five variables constitute a vicious circle that can worsen the prognosis of a patient with anorexia and should be taken into consideration in the management of these patients.

### 3.2. Family and Anorexia

Family context has been highlighted as a key factor in anorexia development and disease management. Hence, the latest literature proposes how parental influence may modulate body image development, since relatives take part in primary socialization, enhancing the first social acceptance. Thus, at the age of approximately two, children usually become conscious of their gender, and they take part in social norms of participation, such as male competitiveness and powerfulness and female beauty or neatness. At this moment, children seek parental agreement, in order to reinforce their personality [43]. Subsequently, as they move towards pre-adolescence, previous authors have pointed out how by 6 years old, body shape gains consideration until 12 years old, relating some non-conformities with aspects related to body size or shape in approximately 50% of surveyed children. Later, children reach adolescence, where physical and social changes become more important, and this crucial period plays an important role in body image development [44]. It has been described how the relationship between relatives and the adolescent may strongly influence the adolescent’s body dissatisfaction, since parents can be a source of different kinds of messages, including critical, sociocultural, or their own ideals about body appearance or eating concerns, which may modulate adolescents’ thoughts and behaviors [43,45]. For example, it has been described how a significant influence on body satisfaction could present in children to whom parents paid special attention in relation to weight control actions. Thus, these messages can increase self-criticism, as well as encourage children to judge themselves or others on the basis of appearance and reinforce the idea that established social and cultural body ideals must be met as proof of acceptance [45,46].

Furthermore, previous authors suggested that adolescents with better parent–adolescent relationships presented a lower probability of suffering from body dissatisfaction [47]. Body disappointment is a complex risk factor which may predispose to anorexia development, maintaining and lengthening illness [48,49]. Previous researchers have pointed out how anorexia patients can be inclined to overvalue their body size, more specifically controversial parts such as hips, buttocks, abdomen and arms [50]. As a consequence, anorexia patients suffer different emotions such as anxiety, disapprobation, guilt and social withdrawal, increasing their discomfort [48]. Thus, it should be considered how powerfully parental influence may affect anorexia progress.

Previous authors supported the idea that neuropsychological deficits, such as cognitive inflexibility and decision-making, could be also observed in first-degree relatives of the patients. Regarding decision-making, it may be that anorexia patients present a dysfunction at this level, since previous research has shown how decision-making was significantly diminished in eating disorder patients, leading to restrictive eating behaviors [51].

Finally, family-based interventions have been established as a useful tool especially in children and adolescents, showing encouraging results in anorexia management and recovery [52,53,54]. Thus, recent authors developed a specific model treatment, which consists of an intervention constituting four phases, which may have a positive effect in families supporting anorexia patients. These phases include commitment and growth of the therapeutic agreement, aiding families to relieve the symptoms of an eating disorder, discover individual and family development issues and finally, to complete treatment and discuss future plans [55]. Naturally, parental influence in this case plays a beneficial role in improving anorexia management, as patients may feel supported by their family members, promoting better strategies in order to decrease the severity and overall health impact of anorexia or even overcoming the disease.

### 3.3. Social Context and Anorexia

The sociocultural context is recognized as an important risk factor for the development and for the specificity of eating disorders [56].

Social performance generally refers to an individual’s ability to interact successfully with their environment (including work, school, social activities, and relationships with partners and family). This is possible by the development of a variety of social skills including verbal and nonverbal gestures, social cognition, and interpersonal performance. Although not a diagnostic criterion, some studies suggest that eating disorders are associated with atypical social and emotional behaviors [57,58,59]. At the same time, other studies highlight the important role of social support [59], social behavior and social inclusion [60] in the successful recovery of adults with ED.

Currently, the media promotes pre-set standards of physical appeal based on thinness. Media coverage interaction with body dissatisfaction, and personality traits could intensify specific behaviors in women that should aid them in achieving an ideal body image, e.g., excessive focus on body image, weight control, increased physical activity. However, the intensification of these behaviors may develop anorexia readiness syndrome (ARS) in women [61]. Such behaviors can be attributed to comparing oneself to unrealistic ideals while setting aside one’s own attractiveness. These behaviors include: increased interest in nutrition, calorie counting, dieting, weight control methods, increased physical activity, excessive focus on body image, emotional lability related to eating and body perception, the desire to control one’s own body dimensions and weight, high competitiveness and perfectionism, as well as the need for control [61].

Several studies have confirmed the significance of sociocultural messages, in highly industrialized countries, in reinforcing the perception that ‘thin female bodies’ are more attractive. Levine et al. [62] found in a group of 10–14-year-old girls that most reported receiving a clear message from fashion magazines and peers or family members that slimness is important and can be achieved through dieting and other methods. Two strong influences on the urge for thinness and altered eating patterns were reading magazines containing information and ideas about attractive body shapes and weight control, and receiving weight/body shape-related criticism from family [62]. In a follow-up study of 428 boys and girls, Bearman et al. [47] found that body dissatisfaction was higher among girls. For both sexes, lack of parental support, negative affectivity, self-reported dietary restraint, body mass index, and eating pathologies, showed significant associations with future increases in body dissatisfaction.

Factors such as body image dissatisfaction, the restrictive pursuit of thinness, the adoption of a perfectionist attitude towards the body, and the development of bulimic tendencies are often indicated in scientific research as predictors of eating disorders. Peer pressure is common to both sexes: A study conducted among adolescent schoolchildren indicated that, while girls talk more about appearance, boys had a higher perception of appearance pressure and teasing. Boys also admitted that they talked to their friends about muscle development more often than girls did about diet. Most researchers, however, focus on the selected risk factor [63].

Additionally, while peers are a major influence in a teenager’s life, another strong social influence for teenagers is social media, media advertisements, and the internet in general. Social media is a part of most people’s lives, especially Facebook, Twitter, Tiktok, and Instagram. These accounts contain dangerous information about body image, eating habits, and physical appearance. Social media is a way for some people to inform themselves and has become a career for others in the form of brand ambassadors or ‘influencers’. However, most research on social media and social networking is somewhat outdated for today’s teenagers, as technology changes so rapidly. Although there is little research on social networking sites, advertisements are a great way to reach different types of people, whether through social media or other media. The internet in general is a tool that many young adults use to define themselves and they can find themselves on sites that are considered “pro-anorexia”, where adolescent girls can find like-minded people [64].

Pro-anorexia websites are virtual spaces where adolescents can exchange ideas about their body image and physical appearance. Uncontrolled use of these websites is a common practice among adolescents, particularly among young women, and is a factor related to eating disorders. According to different studies, looking at images of underweight celebrities is associated with an ideal body image and aspiration to lose weight, and these conditions may promote eating disorders [65,66]. The COVID-19 pandemic influenced media exposure to adolescents due to increased overall media consumption, increased exposure to harmful content related to diet and physical appearance, particularly on social networking sites, and increased use of video conferencing and subsequent self-image exposure while working and/or studying from home. All of these may be associated with an increase in eating-disordered behaviors [67].

Knowledge of the relationship between the use of social media and self-esteem related to eating disorders is necessary to promote prevention of the use of these electronic media, generating skills and providing sufficient information to avoid being negatively influenced by messages broadcast on social media about the ‘Western ideal’ of body perfection [68,69].

### 3.4. Bone and Muscular Implications of Anorexia

Eating Disorders (EDs), specifically AN, are associated with multiple neuroendocrine disruptions and low bone mineral density (BMD) [70,71]. Increased calorie restriction may impede reaching optimum bone mass, especially during adolescence, which can lead to long-term skeletal disorders [71,72]. Dual energy X-ray absorptiometry (DXA) measurements have shown that adolescents with AN have lower BMD and lower rates of bone accretion than normal-weight adolescents of comparable age and maturity [70,71]. Specifically, Z-scores of less than −1 are present in up to 50% of adolescent girls with anorexia nervosa, while Z-scores of less than −2 are present in 11% of these individuals. Despite the fact that there are fewer data on AN in males than in girls, the research that is available is even more alarming, with 70% of boys having Z-scores of less than −1 at age [71]. Thus, osteopenia, lower than normal bone mass and lower BMD, is a frequent and often chronic consequence of AN that causes clinical fractures and an elevated lifetime risk of fracture (Figure 1). Another study reported that only 15% of women with AN in their mid-20 s had a normal BMD across all bone regions investigated, while 55% had osteopenia and 35% had osteoporosis [73]. Moreover, studies in adults have shown lower markers of bone formation and higher markers of bone resorption, which has led to the theory that osteopenia is brought on by low osteoblast activity and high osteoclast activity. Conversely, AN results in a broad decrease in bone turnover indicators during adolescence, a period of significant bone turnover [74,75].

The decreased mineral density phenomenon is amplified when the condition manifests during adolescence and when the length of amenorrhea is prolonged. Although estrogen insufficiency has long been thought to be the principal cause, it is insufficient to explain the condition. Recent research has revealed the critical importance of nutrition-related variables, including leptin and adiponectin (Figure 1) [76,77,78]. However, several mechanisms have been proposed and are probably interrelated, such as growth hormone (GH) and IGF-1 metabolism disorders, hypercorticism, vitamin D deficiency, adipose tissue metabolism disorders and factors involved in adipocyte/osteoblast differentiation. Furthermore, it is important to highlight that significant alterations in body weight and composition, pubertal development, and pubertal hormones like estradiol and IGF-1 that affect bone metabolism occur in anorexic patients [79]. More specifically, the bone-trophic hormone IGF-1 acts on osteoblasts and collagen production to promote bone growth and development. High plasma GH levels and low IGF-1 levels are found in AN patients, which may indicate GH resistance. Furthermore, leptin plays a complex role in controlling bone mass and density. Likewise, peripheral leptin appears to promote bone density whereas it appears to diminish it through a central action [71]. Serum leptin levels are generally lower in AN, and it has been discovered that these levels are related to reduced measurements of fat mass and bone density. The results for adiponectin varied based on the molecular weight fraction of plasma adiponectin that was examined, but some authors observed a rise in serum levels [80].

Thus, these variables accounted for the majority of the variation in BMD in AN individuals with osteoporosis. Additionally, regarding stress hormones, this population has been observed to have high cortisol levels despite regular circadian cycles. A rise in free urine cortisol is commonly observed, with a dexamethasone test reducing hypercortisolism [77]. As is the situation with patients using exogenous corticosteroids, hypercortisolism can lead to reduced BMD via decreased osteoblast activity, which inhibits bone production, and increased osteoclast activity, which promotes bone resorption [78]. Concerning sex hormones, adults and adolescents with AN have lower serum levels of estrogen and testosterone than controls, and estrogen deficiency has been identified as a significant etiological factor for bone loss in this population. Additional potential causes include hypothalamic dysfunction and weight loss, as well as the dysregulation of neurohormones such as GnRH. Moreover, a link has been reported in the literature between BMD, duration, and age at onset of amenorrhea [78,81].

Although bone has received most of the attention at the present, muscular function is also compromised [82]. A consistent low-calorie diet combined with high-metabolic-loading activities may help people lose weight on their own [83]. As a result, patients’ body composition profiles experience significant changes in their bone mineral, water content, relative body fat, skeletal muscle mass, and fat mass. Depletion of fat free mass, which contributes 15 to 45% of the total body weight decrease, is associated with diminished muscular performance [84,85]. The initial therapy objective for people with AN is to reduce calorie deficit by increasing total calorie intake, which will then lead to weight restoration. Current epidemiological evidence suggests that across many patient populations, muscular size and strength—not necessarily weight—predict lifespan, quality of life, and mortality [86]. However, regaining muscular mass and strength has been generally ignored in the treatment of eating disorders. Maintaining muscle mass requires a fine balance between protein production and protein breakdown. In this regard, mTOR (mammalian target of rapamycin) promotes protein synthesis in a wide sense. Activation of mTOR moderates the stimulation of downstream effector proteins (4EB-P1 and S6K), thus facilitating the translation of mRNAs into polypeptides for protein synthesis [87]. mTOR can be activated by a variety of stimuli, with resistance exercise and dietary protein consumption being two important activators in humans. mTOR inhibition frequently results in diminished protein synthesis and diminished muscle mass [88] and a resulting decline in muscle strength. Despite the fact that our understanding of the mTOR regulatory mechanism has allowed us to better comprehend the pathophysiology of malnutrition over the past two decades, numerous important concerns remain unanswered. A recent review focused on the mTORC1 signaling pathway as an essential energy sensor, which plays a crucial role in the regulation of whole-body energy balance, centrally and peripherally [89]. Although our understanding of these nutritional and hormonal adaptive mechanisms in AN that maintain vital functions during this severe form of wasting has been increased, the central role of the mTOR system in the hypothalamus and subcortical areas in the regulation of energy balance in AN patients remains unknown and these interconnected but distinct systems may change during AN or after weight recovery [89].

Considering the possible hereditary inheritance of eating problems, in skeletal muscle the genes that regulate mitochondrial functions are also determined. It is crucial to consider how AN is influenced by genetic elements. Consequently, DNA was taken from two large ED-affected families, and whole genome sequencing, linkage mapping, or whole exome sequencing were used to identify segregating variants associated with the disease [90]. Twenty members of the first family, spanning three generations, were found to have a disease-correlating uncommon nonsense mutation in the estrogen-related receptor α (ESRRA) gene. In the second family, a nonsense mutation in the histone deacetylase 4 (HDAC4) gene was linked to the illness after eight people over four generations were examined [91]. Moreover, it was discovered that ESRRA and HDAC4 interact both in vivo in the mouse brain and in vitro in HeLa cells [91]. Assessed in greater detail, ERRs are members of a small subfamily of nuclear receptors termed NR3B, which has three members: estrogen-related receptor alpha (ESRRA/NR3B1), beta (ESRR β/NR3B2), and gamma (ESRR γ/NR3B3) [92]. Specifically, ESRR α and ESRR γ are predominantly expressed in metabolically active tissues that use fatty acids as fuel (such as the heart, brown adipose tissue (BAT), brain, gut, and liver) [93]. ESRR α controls mitochondrial activity, biogenesis, turnover, and lipid catabolism [94,95]. Additionally, it regulates appropriate physiological and developmental muscle and bone function [92]. It has a documented involvement in energy balance and metabolism, is increased in peripheral tissues by exercise and caloric restriction, and is a transcriptional target of the estrogen receptor [91,96,97]. However, one association research study evaluating 182 potential genes excluded ESRR α and HDAC4, but included PGC-1, the ESRRA coactivator [98]. Thus, mutations altering the functional relationship between the transcription factor ESRR α and the transcriptional repressor histone deacetylase 4 (HDAC4) could be linked to the development of eating disorders (Figure 1). Although it is believed that eating disorders (EDs) develop from a complicated interaction between genetic susceptibility and environmental risk factors, studies have failed to discover specific genes that predispose to the development of an ED.

### 3.5. Microbiota and Anorexia

Among the different biological parameters that potentially constitute and influence the patient with AN, the gut microbiota has recently gained a lot of attention. It is logical to think that if its composition and health is largely due to the composition, richness, variety and quality of the food we eat, in a patient with AN in whom malnutrition is present, alterations will be observed.

The human host’s intestinal microbiota is an ecosystem made up of bacteria, archaea, microeukaryotes (including fungus and protozoans), and viruses that coexist in harmony. In the human microbiome, bacteria outweigh archaea and microeukaryotes. There are a total number of around 3.8 × 10^13^ bacteria in the 70 kg “reference man” [99], which is on par with the amount of human cells in the entire body. Firmicutes (60–65%), Bacteroidetes (20–25%), Proteobacteria (5–10%), and Actinobacteria (3%) make up the majority of the bacteria in the gut [100]. Even though these phyla are uniform, the hundreds of bacterial species found in the microbiota differ substantially across people. The functional gene profiles of bacteria are fairly comparable between people, in contrast to the variation among bacterial species, which may indicate the presence of shared fundamental activities [101]. Digestion and fermentation of nutrients, particularly carbohydrates and amino acids, as well as the creation of important metabolites such as short chain fatty acids are tasks carried out by gut microbes.

#### 3.5.1. Diversity and Microbial Metabolites in Anorexia Nervosa

Recent reviews suggest that there is diversity and composition differences in the intestinal microbiota in patients with AN in comparison with healthy controls. Borgo et al. [102] showed that the families of dominant bacteria in the patients with AN are Bacteroides, Firmicutes and to a lesser extent Actinobacteria, Proteobacteria and Verrucomicrobia, a composition similar to that presented by the healthy groups. However, Mark et al. [103] stated that the Firmicutes and Bacteroides concentrations were decreased in patients with AN. Both studies ensured that patients with AN had elevated concentrations of Actinobacteria, Proteobacteria, and Enterobacteria when comparing them with healthy controls. Another study by Morkl et al. [104] supported the idea that patients with AN also had elevated levels of Coriobacteriaceae. On the other hand, patients with AN have also shown decreased levels of Ruminococcus and Roseburia, both butyrate producers.

However, differences in the microbial compositions of the clinical subtypes of AN, either restrictive or purgative AN, have also been analyzed by authors. However, restrictive or purgative AN differ in their feeding behavior since the first is characterized by the severe restriction of caloric intake while the second occasionally eats large amounts and is often accompanied by vomiting [103,105]. Authors concluded that there were no significant differences between the subtypes in terms of quantity.

Numerous variables, including food, affect the quantity and composition of the gut microbiota. In fact, both short-term and long-term dietary modifications can cause detectable microbial alterations [106]. There is evidence that some situations, such as acute starvation, might lead to an imbalance in the gut microbiome (also called dysbiosis). The detection of particular bacterial and archaeal species in feces from AN patients in comparison to healthy persons using quantitative PCR provided the first data indicating that the intestinal microbiota from AN patients may differ from healthy individuals suggesting dysbiosis [107]. According to these findings, 9 AN patients had higher levels of the methanogen *Methanobrevibacter smithii* (*M. smithii*) in their gut microbiota than did 20 healthy normo-weight subjects. Indeed, it is *M. smithii* and other methanogens that increase microbial fermentation and the conversion of nutrients into energy by breaking down excess H2 in the gut into methane. The authors suggest that the higher levels of *M. smithii* may thus be an adaptive response to optimize energy extraction from the extremely low calorie diet that anorectic individuals are consuming [107].

Constipation, a common functional intestinal condition seen in AN patients, might potentially be connected to this rise in *M. smithii*. In fact, an increase in methane-producing bacteria has been linked to patients with constipation, and more specifically, patients with C-IBS (irritable bowel syndrome with a constipation predominance) [108]. There is currently some data that suggests methane decreases gastrointestinal motility and may thus contribute to constipation.

The dysbiosis that occurs in AN patients affects not just Eubacteria and Archaea, but most likely the entire gut ecology, which includes viruses and eukaryotes. Research concentrating on the gut microeukaryotes of an AN patient found a decline in fungal diversity and discovered four species previously unknown in the human gut. Again, these findings must be extrapolated to other AN patients and supplemented by a large-scale investigation of the variety of viruses and eukaryotes in these individuals.

#### 3.5.2. Psychopathology in Anorexia Nervosa

Breton et al. [109] demonstrated that *E. coli* produces CIpB, and this is in turn an anorexigenic protein, which could establish a relationship between the Enterobacteriaceae and AN, that is, a gut–brain communication. In addition, the caseinolytic protein B (CIpB) is associated with melanocyte stimulant (MSH), a hormone that is involved in signaling satiety and anxiety characteristic of eating disorders.

These findings supported the results obtained in the studies carried out by Borgo et al. [102] and Kleiman et al. [110], where they analyzed the association between the microbial composition and psychopathology of AN.

Depression and anxiety were the two most common symptoms in these patients, highlighting the personal dissatisfaction and social insecurity in patients with AN. Authors observed a negative correlation between bacterial species, mainly Clostridium concentrations, and butyrate levels with the symptoms of depression. These results point to the influence of the microbiota in the intestine in the regulation of mood in patients with AN [110].

#### 3.5.3. Gut Microbiome Rehabilitation

Authors confirmed that the development of microbial diversity is linked to weight gain after treatment; although authors show an improvement in the composition of the microbiota that was considered a reference after comparison with the healthy group [103]. It was shown that the microbiota of AN patients after treatment had a higher quantity of Firmicutes, Bacteroides and Ruminococcus, which could be explained by the diet prescribed during the rehabilitation that was characterized by being rich in fiber. Regarding the plasma levels of SCFA, there was no recovery after nutritional rehabilitation. The study carried out by Kleiman et al. [110] and Prochazkova et al. [111] confirmed the increase in microbial richness in patients with AN after treatment, however, there had been no complete restoration of the microbiota when compared with the control group.

In another study the microbiota of patients with recurrent AN was analyzed, after fecal transplantation authors reported results of increased microbial diversity. These findings could suggest that gut dysbiosis is one of the causal factors in the etiology of patients with AN [112]. Gastrointestinal symptoms characteristic of patients with AN may be affected by the modifications produced in the diet or by the composition of the intestinal microbiota. Therefore, nutritional rehabilitation is key to mitigating gastrointestinal symptoms such as constipation.

Physical activity, which is reduced in patients with AN to avoid energy expenditure and thus promote weight gain; could be beneficial in a controlled way for nutritional rehabilitation since two studies carried out by Morkl et al. [113] and Speranza et al. [114] demonstrated the benefits of physical exercise on the composition of the intestinal microbiota through the study of athletes. In summary, from these studies it is shown that patients with AN present an altered composition of the microbiota that allows the establishment of a gut–brain association; and postulates the possibility that the microbiota is the cause of the development of this disease, therefore suggesting that an analysis of the intestinal microbiota at the initial evaluation of AN could be a useful tool for subsequent nutritional rehabilitation.

### 3.6. Dental Health in Anorexia

There has been an increasing focus on the poor physical health of persons with mental illness, but less attention has been paid to dental health, despite the fact that it is an essential component of physical health [115]. Moreover, painful, ugly dentition or ill-fitting dentures can worsen social disengagement, isolation, and low self-esteem, as well as create speaking and eating difficulties. In addition to this, there is a link between oral illness, coronary heart disease, stroke, diabetes, and respiratory disease. However, even in nations with universal health care coverage, dental care is not completely covered [116]. Those with mental illness, particularly severe mental illness, are at increased risk for dental health problems because of poor diet and oral hygiene; the high consumption of sugary drinks; co-occurring substance abuse, such as cigarettes, alcohol, or psychostimulants; and finasteride [117]. To reduce barriers to care, whether they are psychosocial or financial, more coordination between mental health clinicians and dentists is essential [116].

The oral cavity may be the principal site of involvement in endocrine, renal, gastrointestinal, cardiovascular, hematological, autoimmune cutaneous, and psychological illnesses. EDs are psychosomatic disorders with complex etiologies and aberrant eating behaviors [118]. Brain alterations in AN were discovered decades ago, with the most prevalent result being a loss in grey and white matter that correlates with the severity of malnutrition and is typically reversible upon recovery (Figure 2) [119]. In many situations, the oral cavity may be the only region where eating disorders occur. The medical complications linked to these disorders (dehydration, electrolyte abnormalities, abnormal heart activity, gastrointestinal complications, endocrine disorders, osteopenia, and increased risk of fertility problems) have been well described in the medical literature for many years [120]. Nevertheless, the effects of eating disorder behaviors on the teeth and oral tissues were not recognized until more recently [121,122].

One of the most severe eating disorders in the world, restrictive type AN, still has an unfavorable prognosis as 1.2–2.2% of girls and women will develop full-blown AN throughout their lives [123]. Frequently, the onset of an anorexic eating disorder occurs as young as 12 years. According to Roberts and Li [124], the extremely low self-perception and self-esteem that many sufferers of AN and BN experience may contribute to their poor oral hygiene and increase in dental disease. The critical nature of oral health at this age may be attributed to the phase of development during which permanent teeth mineralization and periodontal tissue are formed [125]. Any oral imbalance may have permanent effects on their oral health in the future. In this regard, research including adult participants revealed significant dental caries, erosive tooth wear, and loss of periodontal health [116]. Comparing the oral cavities of anorexic young patients to those of the control group in a recent case study, the most significant findings were the presence of dental caries and poor oral hygiene in conjunction with gingival bleeding. Dental examination revealed that 37.6% of AN patients were affected by dental caries, compared to 11.7% of the controls [126]. In this regard, few dental analyses were based on an adult patient population [127], the majority consisted of a small number of cases ranging in age from 12 to 18. However, the perioral tissues, oral mucosa, teeth (such as dental erosion and dental caries), periodontium, salivary glands, and temporomandibular joint are among the soft and hard tissues that are affected in both populations [128]. Salivary adenopathy, hyposalivation/xerostomia, dental erosion, periodontitis, and soft tissue disorders are also among the common oral signs of ED [129] (Figure 2).

Similarly, the Decayed, Missing, Filled teeth (DMFT), dental caries score of the AN individuals were considerably greater than that of the control group [130]. However, previous research based on the Basic Erosive Wear Examination (BEWE) index examination found that AN patients had a lower risk of tooth erosion than BN patients. Notably, young people with AN experienced infrequent vomiting inducement [131]. Nevertheless, the AN subgroups with purging episodes had worse dental erosion results [126]. Moreover, recent studies reported that 47.3% of AN patients declared suffering from bleeding gums after tooth brushing as the most prevalent oral health symptom, 37.5% of patients also regularly complained of tooth hypersensitivity [132].

Patients’ eating habits are influenced by their intense fear of gaining weight, preoccupation with weight, denial of their current low weight and its negative impact on their health. Oral signs of EDs are influenced by the patient’s diet, level of oral hygiene, frequency and length of dysfunctional behaviors, ability to induce vomiting, and usage of drugs. In addition to vomiting, many factors influence the BEWE score. The role of an acid diet may be significant; foods and beverages tend to decrease the pH of the oral cavity, particularly in those with tooth decay. It is hypothesized that a diet low in pH, particularly beverages, and excessive physical activity may decrease salivary buffer capacity at this age and encourage the loss of non-caries dental tissue [133]. Poor motivation to maintain proper dental hygiene may also be influenced by a challenging life circumstance and the linked apathy, sad mood, psychomotor drive, and suicidal tendencies, according to our observations and research [134]. Other studies found that a diet deficient in protein, vitamins, and unsaturated fatty acids contributes to metabolic and biochemical abnormalities that result in an imbalance between oxidants and antioxidants [135]. Therefore, the persistent state of malnutrition, particularly in extremely severe AN, impacted the likelihood of medical problems and fast gingival inflammation [136]. In order to give patients an interdisciplinary approach to care for the whole person, it is crucial to include an oral health program focusing on enhancing self-image through dental education [137]. Specifically, an oral health education protocol that is standardized and included in the normal curricula of eating disorder treatment facilities should be developed by taking into account the oral health knowledge and self-image beliefs of eating disorder patients in connection to their smiles.

## 4. Non-Pharmacological Interventions in Anorexia Nervosa

Following a thorough assessment of each of the non-pharmacological strategies that could most effectively aid those who are suffering from anorexia. Table 1 below presents some of the most important references from this review.

### 4.1. Nutritional Interventions

Treatment of AN is based on how the disease presents itself and the severity of it, therefore every patient’s treatment should be individualized [148]. Nutrition is a very important part of managing this disease, since when malnutrition is present, the psychological treatment of the patient is not possible. A malnourished person cannot mentally center on the problems and solutions related to their disease [149].

Dietitians are in charge of the differential diagnosis with other disorders, establishment of the most appropriate nutritional treatment, evaluating the need to integrate other health care providers into the treatment, assessing the need for hospitalization, evaluating the need for prescribing protein and caloric supplements, assessing the need for using a nasogastric tube or parenteral nutrition, follow up with the patient’s recovery, involving the family members in the treatment of the disorder [148], and obtaining a food history, which can be more useful than lab tests in determining nutritional deficiencies [150].

The first step in the treatment is the clinical history, and what is usually common is that even when the physical aspect of the patient is very concerning, they are not aware of the disease. These patients usually do not go willingly to counseling and the first step is to establish a good relationship with them. The dietician should ask about other health complications that might be present such as the presence of vomiting, the intake of laxatives or diuretics, drug consumption, and the frequency of physical activity. The dietician should obtain information about what foods they intake and what are the forbidden foods for them, also the quantity and how such foods are consumed (e.g., chopped into small pieces) [151].

The physical examination is the next step. Usually, the most notorious sign of the disease is the loss of subcutaneous fat. Other symptoms can be a decrease in body temperature, hypotension, and lanugo. When the weight is determined, the nutritionist should make sure that the patient did not try to manipulate the result by ingesting excess water, or by hiding certain objects in their clothes.

The primary aim of the nutritional treatment should be to restore the body weight back to normal, treating all non-acute physical complications, educate the patient on healthy eating habits, modify any other associated disorders such as purging or binge eating disorders, improve the perception of anxiety and hunger, obtain family support, and prevent any relapse [152,153]. In teenagers and children, it is important to restart growth and development. It is important to restore at least the patient’s minimal weight, but in order to determine this weight, other factors should be taken under consideration, like family weight history, patient’s growth and development rate, patient’s ponderal evolution, sex, and age [154].

The energy requirements should be calculated based on the patient’s actual weight instead of the patient’s ideal weight, this calculation will also depend on the patient’s degree of malnutrition [138]. It is not recommended to use the usual formulas to calculate these requirements since they tend to overestimate the patient’s needs at the beginning [154] and later in the treatments, they can underestimate them [155]. Usually in patients with AN an increase of 0.5 kg a week could be set as a reasonable target [153,156]. In the first six months of treatment, patients suffering from this disorder require more caloric intake than healthy patients in order to maintain their body weight [138]. Weight is important for evaluating patient progress, but this should not be the only goal of the treatment, outcomes have more importance than weight gain to determine the progress of the patient [157]. Additionally, water retention is very common when refeeding starts, and this could make the weight of the patient increase, and give a false result [158].

There is no evidence regarding what is the ideal food or meal for this disorder, the diet should be based on a healthy food intake pattern [159], and the food quantity should be determined by the dietitians, depending on the calculated energy requirement [154]. If for any reason the patient cannot reach the calculated energy requirements, nutritional supplements should be prescribed [138]. One out of three patients with AN present with vitamin deficiency [149]. This is due to lack of vitamin intake or due to the fact that the food restriction caused thyroid hormone abnormalities which impede the metabolism of riboflavin [160]. This is an important phenomenon that nutritionists must take into consideration [149]. Osteopenia is a common complication of the disease, intake of vitamin D and calcium are important, and even with the patient’s recovery a lower bone density and a deficiency in bone minerals can still be present [161].

Nutritional education is extremely important in the treatment of AN, since its aim is to modify the eating behaviors to transform them into more healthy patterns [162]. The appropriate education could lead patients to develop skills that allow them to choose healthy foods and maintain a positive attitude [139,163]. Nutritional education has several objectives, among which are: to improve the behavior and relationship with food, reach an adequate number of meals a day, improve the energy intake of micro and macro nutrients, improve the alimentary pattern to achieve the recommended intake of the basic food groups, and to clarify myths and errors obtained as a result of inadequate information sources.

Patients suffering from this disorder usually have a deep understanding of the caloric content of every food, but they do not have any nutritional knowledge, which will help them improve their health, once the fear of gaining weight is overcome. Patients are usually misinformed about which foods are healthy and unhealthy, and their information often comes from nonscientific sources and creates fear and confusion [135,164]. This is why nutritional counseling is so important, and it should focus on controlling these abnormal behaviors around food and body weight to avoid any future relapse [143,165]. This can only be achieved by making the patients aware of the problem and instructing them about the benefits of changing these patterns [166]. Other activities like preparing meals or grocery shopping can also play an important role in the patient’s recovery and in establishing a good relationship between the patient and food [149]. Nutritional education plays an exclusive role in the overall treatment of AN [149] and should be given by dieticians that have extensive knowledge of eating disorders. It is important to take into consideration that changes generated by the education in nutrition are nonlinear, meaning that patients usually experience setbacks for different reasons and need to start over [135].

### 4.2. Physical Activity Interventions

The relationship between physical activity and anorexia is paradoxical because it has been associated with negative and symptomatic effects on the health of anorexia patients, or more recently, structured and supervised exercise has been proposed as coadjutant therapy in patients with anorexia because it might improve eating disorder symptomatology, strength and muscle function and vital signs [167]. On the one hand, excessive physical activity has been identified in around 80% of patients with anorexia as a tool to produce a greater daily energy cost and weight loss and to influence their body shape [168,169,170]. In addition, as exercise has demonstrated its effect on negative affective states such as stress, depression or anxiety, patients with anorexia used excessive exercising as a strategy to alleviate the aforementioned emotional, mood, behavior and other eating disorder symptoms (e.g., body dissatisfaction, weight preoccupation) [171]. Moreover, physical activity and exercise are linked to increased endorphin levels which produce euphoric and analgesic feelings, creating a relaxed psychological state and playing a key role in addiction [172]. For these reasons, excessive exercise has been associated with a negative impact on clinical outcomes (e.g., muscle mass decrement, osteoporosis, electrolytic imbalance) and poorer quality of life [168,173], increasing physical complications and dropouts during the treatment and the risk of relapses [167]. Therefore, physical activity should be programmed and well controlled during the different phases of the pathology because compulsive and non-supervised exercise during the acute phase and during the hospitalization phase of the disorder can affect the therapeutic targets [174].

On the other hand, current evidence does not report adverse effects of supervised exercise interventions on weight loss and other clinical outcomes in patients with anorexia [167]. This finding is also supported by a previous systematic review on the topic [140] that found that supervised exercise training did not modify body composition (body fat, body weight and body mass index) but can have benefits in strength and psychological outcomes in patients with anorexia, concluding that supervised exercise training programs are safe and must be included in the treatment therapies of this eating disorder. Another systematic review [167] corroborates previous affirmations, suggesting the inclusion of a supervised physical exercise program as a coadjutant therapy during the usual treatment of anorexia may produce a positive impact on symptomatology. Specifically, supervised strength training may be a promising tool in the treatment of patients with anorexia. Previous studies showed beneficial effects of resistance training on muscle mass and strength values in patients with anorexia [85] which are two outcomes that are clearly related to the risk of mortality, other comorbidities and quality of life [175,176,177]. Moreover, the inclusion of two training session of 20 jumps daily [178] promoted health benefits in vital signs in patients with anorexia. Interestingly, aerobic exercise and flexibility has been also used as adjunctive therapy in anorexia treatment [179,180] reporting no negative effects on weight gain. In this way, yoga has been proposed as a tool to produce benefits in symptoms such as anxiety, depression, or general eating disorders in patients with anorexia [180]. Individualized yoga treatments may reduce global eating disorders psychopathology and improve body image concerns [181]; yoga has been proposed by experts as an adjunctive therapy that should be included in future guidelines for anorexia treatment, in order to manage symptoms of anorexia (anxiety, depression or excess of exercise) [182]. Therefore, although substantial evidence supports the implementation of supervised exercise programs to treat anorexia [85,140,141,183,184], knowledge of the exercise characteristics (frequency, duration, intensity, type of exercise) to obtain an optimized program is scarce and further research is required. Regarding the aforementioned exercise characteristics, some remarkable aspects have been analyzed in recent studies. For example, the effect of high-intensity training on anorexia treatment has been explored in a previous study [141] and it has been associated with a better nutritional status. This previous study adapted the exercise intensity to the patient’s clinical status and the exercise program was individualized and continuously monitored during the rehabilitation process. Furthermore, it has been previously suggested that the calorie intake needs to be adjusted in patients that perform physical activity or exercise [85,183]. Therefore, the individualization of the supervised exercise program based on strength and flexibility exercises may determine the effectiveness of the intervention.

### 4.3. Psychological Interventions

The disorders found in the classification of Eating Disorders according to the Diagnostic and Statistical Manual of Mental Disorders (DSM-5) are very complex pathologies, which have multifactorial etiology and mainly affect young people between the ages of 12 and 25 years. However, although there is currently a higher tendency of appearing at younger ages and at older ages, and entering the adult stage. Regarding the prevalence of these disorders, it is estimated that approximately 4.2% of young people between 12 and 24 years of age have an eating disorder, approximately 3% have unspecified disorders, almost 1% have bulimia and 0.3-0.5% have anorexia [185]. On a general level, it is estimated that more than 3 million people a year have their health significantly worsened by these disorders, which increases the number of people who must live with some kind of disability due to poor nutrition [186,187].

In terms of comorbidity, the most frequently associated disorder is depressive disorder, showing up to 24% in cases of anorexia, and 65% in cases of bulimia and binge eating disorder. As for anxiety, we found that at least 10% present depressive symptomatology in AN, and between 45 to 50% in the case of BN and binge eating disorders [18,188]. The clinical manifestations of the most common disorders, anorexia and bulimia and binge eating disorder are different, although they present as a nuclear symptom the fear of having a weight above that which the person has idealized [189]. Associated with these symptoms we also find an irrational fear of gaining weight, body dysmorphia and comorbidity with other serious disorders such as anxiety, depression, substance use or personality disorders [190]. Recent studies along these lines also show a predisposition for these patients to present self-injurious behaviors and suicide attempts, which is an element that causes these disorders to have a high mortality rate. Suicide is the second leading cause of death in these disorders, after medical problems associated with malnutrition [191].

It is known that there are some risk factors that can function as precipitators of the onset of the disease. Likewise, these factors can explain the maintenance of the pathology and the difficulty in recovery [192]. Among the most frequent in the psychological and cognitive area we find low self-esteem, the previous presence of mood disorders, as well as some more physiological elements that are associated with the body changes that occur during adolescence and that can confuse the person and determine a high dissatisfaction with his or her physical image. Social factors often include the presence of different family and educational problems and rigid mental schemes in relation to body image and success patterns that include beauty ideals [193,194].

In this sense, the emergence of the SARS-CoV-2 pandemic in 2019 posed a great challenge for these patients, since those elements precipitating the disease were aggravated in a highly restrictive environment, increasing levels of anxiety, stress, and depressive symptomatology, at the same time as a significant decrease in quality of life [195]. All this led the population to reduce social contact and thus increase isolation, the perception of loneliness and concern about how to cope with an exceptional situation. In addition to this, health care was blocked worldwide due to the massive contagion and the growing number of deaths, which left mental health care, among others, in second place, with the interruption of assistance to these patients who did not have alternative ways to adequately address the symptoms associated with the pathology presented [196,197].

The treatment of eating disorders should be approached from different areas and using the resources best suited to the individual situation of each patient, which will be determined according to the level of severity of the disease and the determining factors that are favoring the presence of this pathology [198]. Therefore, the approach must always be multidisciplinary, involving the joint work of the psychiatrist, the endocrinologist, the nurse, the nutrition specialist, and the psychologist [66].

From a psychological perspective, the most widely used therapy now is cognitive-behavioral therapy, as it has shown the most empirical evidence in the last decades and because it has numerous studies that support it [199]. In addition, it is a therapy that will allow the mental health professional to establish the principles that support the disease, such as cognitive schemas, distortions about body image, food, and weight [200]. Once these bases are established and the patient’s functioning is identified, cognitive restructuring and exposure therapies can be applied to those irrational ideas and fears that are underpinning the pathology [201,202]. In addition, this therapy is useful in disorders that frequently occur along with anorexia, bulimia or binge eating disorder, such as depressive symptoms, obsessive symptoms, anxiogenic symptoms, personality disorders and even some organic syndromes.

Once the main treatment objectives have been established, it is essential that a good definition is obtained of the patient’s clinical history and the relationships that have been developing with food, such as restrictive or purgative behaviors, compensatory behaviors such as excessive physical exercise, and other mechanisms that the patient may have acquired to avoid food. This will allow joint action by health professionals to address recovery [203].

Cognitive behavioral therapy will implement tools aimed at cognitive restructuring to enable the patient to modify those mental schemas that have become maladaptive and have become part of the person’s reality [204]. The objectives include making the patient aware again of the dysfunctional patterns of restriction, compensation and cognitive inflexibility derived from rigid thoughts and ideas regarding food [205]. This is aimed at improving self-image perception, self-concept and ultimately improving self-esteem, as well as an adequate determination of the feelings and thoughts associated with body image [206].

All this should be done progressively, setting milestones that the patient can achieve little by little. Techniques are used to encourage self-control during mealtimes, as well as techniques to increase motivation to change [207]. In addition, it will always be considered that the patient must be prepared for possible relapses, so it is important that they are part of the recovery process and must be prepared to face these moments [208].

Although this therapy is the most effective, not all patients respond as expected, and therefore there are other psychological treatment choices that can be implemented [209]. Among the most widely used, we find schema-focused therapy, the basis of which is to address those dysfunctional schemas that have been created in early stages and that may be resistant and therefore perpetuate the acquired habits [210]. This therapy unifies several psychological models and is based precisely on addressing those cognitive schemas that are sensitive to change and their motive in each person [211].

On the other hand, we find cognitive behavioral therapy focused on appetite, which is based on the analysis of food from the organic point of view, teaching the patient to listen to the signals that the body sends at every moment, both hunger and satiety, so that the person can be able to recognize and ignore the wrong signals in relation to eating behavior [142,212].

Acceptance and commitment therapy (ACT) is showing great empirical evidence in the treatment of these disorders, and is oriented to satisfaction with body image [213]. This therapy currently maintains the great support of the scientific community because it focuses on those behaviors that are valuable for the person, understands and collects the suffering as an inherent part of the human being, addresses relapses as an essential path in the healing process and emphasizes the importance of personal values individually, since these values will be part of the behavior that the patient will implement during the therapeutic process [214]. This therapy is not oriented to the elimination of disruptive thoughts but gives importance to the patient’s flexibility to accept them without a negative emotional bond [215].

In any case, psychological treatment in these disorders is fundamental, since the person needs to be aware of the patterns, ideas and thoughts that are maintaining the disease and that must be modified for recovery to be possible.

### 4.4. Psychosocial Interventions

These disorders have experienced a significant diagnostic increase in recent decades associated with the extreme preoccupation with thinness and body weight. It is very interesting to see how the incidence of these disorders is centered in industrialized societies where aesthetic stereotypes are determinant. In the last twenty years, the incidence and prevalence have accelerated in Western countries, with practically no appearance in underdeveloped societies [186].

Psychosocial factors are determinant in the genesis of eating disorders (ED), as they contribute to the maintenance of the pathology and also to the increase in its appearance in recent decades [216]. Among the most important we find an increase in the perception of an ideal body image as a basis for success in society, and the loss of other cultural and moral elements, changes in eating habits and a lack of commitment to the family. The latter seems to be a fundamental element since the affected person must find in his or her family environment a safe and reliable communication space, which allows dialogue and the intervention of the family as a protective and unconditional support factor [217].

Regarding psychosocial intervention, we are referring to the actions to be implemented to prevent or reduce the occurrence of eating disorders. That is why we work in different stages, the first being the most important; the prevention stage [218]. The second level of action is applied when the pathology is already present and focuses on the identification of symptoms and signs that may be favoring the maintenance. The third level in this type of intervention focuses on preventing relapse when the patient has achieved partial or total recovery and avoiding the appearance of other associated organic pathologies that may increase the risk of mortality [219].

Therefore, the first level is the most important and must be approached from different points of view, including schools, the media, fashions, nutrition and other aspects such as physical activity [220].

The perception of body image is a key factor in these pathologies. In today’s societies this aspect is overvalued and is informed by the media with a vision of the perfection of the slim body [221]. The cult of slim, beautiful women and men is promoted, and is accompanied by attractive messages about health and the improvement of well-being, the achievement of success in the workplace and in the social and personal spheres. As for food products, those that help to reduce or control weight are advertised with the idea that the person who consumes them will improve their quality of life and their position in society. Studies along these lines show the harmful effect of this type of advertising on populations vulnerable to these pathologies, such as young adolescents [222].

It is therefore essential to control advertising, intervene in advertisements that defend the beneficial properties of consuming products that help to reduce weight or not to put on weight. It is necessary to modify the message and the image models used so that the information reaching society is not based on the idea that thinness is synonymous with happiness and success [223].

The acquisition of adequate nutritional habits is the strategy followed in the psychosocial intervention of eating disorders. It is important that these people learn to maintain a healthy diet, which favors education and food awareness [224]. Work is done by education on the products that are supposedly fattening and that become forbidden food [144]. This type of behavior is usually the antecedent of a disorder since foods are restricted or begin with the realization of compensation behaviors before the impossibility of not eating them [224]. That is why it is important that prevention is approached with the learning of healthy eating behaviors, but which are not restrictive or impossible to maintain, which can be diets that can be modified over time and adapted to everyone and their personal situation [225].

An additional issue to be addressed is the facilitation of strategies to deal with new trends or fashions that have a direct impact on the groups at risk of these disorders. The models that take hold as prototypes of styles that are called harmonious, do not have to be valid for all people [226]. However, the tendency of younger people is to follow this fashion even if it does not fit their body measurements. This causes suffering and frustration and can be a precursor to risky eating behaviors [227,228]. This is why it is important to train young people in coping skills in the face of the amount of information related to the new fashions that are appearing. Studies along these lines show the negative impact of fashion standards on young people and how the lack of coping skills leads to negative thoughts about self-image and self-concept [229].

Psychosocial intervention strategies should be applied mainly when risk factors are identified during adolescence. We know that these factors are essential for the appearance of the core symptoms of eating disorders, and among the most important, the following should be noted as determinants: the existence of some type of abuse during childhood, family presence of eating disorders or obesity, family presence of personality, mood or other mental disorders, rigid nutritional habits, body dysmorphia or dissatisfaction with self-image, the presence of excessive physical activity routines, low self-esteem. A systematic review by Linardon et al. (2019) showed that the identification of these elements is essential for a correct multidisciplinary intervention [230].

An early recognition of any of these factors associated with changes in eating routines will allow the application of an intervention that addresses the most dysfunctional aspects and is supported in the environments where there may be more affectation, such as the family, school, or work [144,231]. Although the first therapeutic objective will always be for the patient to regain his or her ideal weight, the psychosocial intervention should be oriented towards the restoration of healthy habits to avoid relapses, since these are a poor prognostic element in the course of the disease [232].

### 4.5. Physical Therapy Interventions

It is recommended a multicomponent intervention program for the treatment of anorexia [146], where physiotherapy could collaborate in the treatment of the distorted body experience by therapies focused on behavior, attitudes and perception [233]. Moreover, physical therapy can also contribute to the observed excessive use of physical activity but this topic has been explained previously. To successfully accomplish these therapy targets in patients with anorexia, physiotherapy has a wide range of therapeutic techniques. They can be applied in a group or in an individual manner.

Among these possible techniques, postural exercises are recommended to be included in the treatment because a relationship exists between the duration and severity of the eating disorder and the possibility of reporting lower back pain [234]. In this way, patients with prolonged anorexia present postural disorders mainly due to the low muscle tone and weakness that promote posture compensations (e.g., scoliosis) and lower back pain [235]. Therefore, the inclusion of strength exercises, mainly core exercises, and postural control and stability tasks can produce a greater tone in the hypotonic muscles, reducing postural abnormalities and also increasing self-esteem. Moreover, to normalize the tone of hypertonic muscles, the physiotherapeutic treatment of anorexia includes stretching exercises. Previous evidence supports the effectiveness of these low intensity exercises [236], showing improvement of health markers and quality of life without negatively impacting weight or body mass index recovery after the exercise program. Moreover, a recent study found similar findings using a combination of stretching and resistance exercise where the stretching exercise of the major muscles were performed at the end of the training session [237].

Additionally, in the treatment of patients with anorexia, massage has demonstrated its effectiveness in attenuating many of the symptoms associated with the eating disorder. Specifically, massage reduces stress and anxiety levels and decreases stress hormones (cortisol) [146]. Moreover, massage decreases the level of body dissatisfaction [235]. The main mechanisms that explain the positive effects of massage are linked with the greater parasympathetic arousal due to the catecholamines and cortisol decrement after the massage [238], the increase in vagal tone and the higher promotion of serotonin and dopamine which can improve mood state [239]. The techniques recommended include passive mobilization of limbs and legs, patting, subtle touching and kneading to increase the relaxing feeling [235]. In addition, relaxing and activating back or leg massage is suggested as the most common massage approach for patients with anorexia [235].

Regarding, relaxing feelings, relaxation exercises and also acupuncture are recommended to treat anorexia symptoms because they reduce anxiety and stress (perceived and cortisol) [147,240]. Among the different techniques which have been demonstrated in their effectiveness in patients with eating disorders (i.e., Jacobson’s progressive relaxation, biofeedback, autogenic training, yoga and mindfulness) [180,235,241,242], the physiotherapist should choose the most relevant method according to the patient’s phase, severity and characteristics within the disorder. The relaxion exercises may be combined with breathing exercises in order to reduce breath frequency with a concomitant increment in the amplitude of abdominal respiration and lengthening of the duration of the expiration phase [235]. These breathing exercises improve breath control and the self-perception of the body by the patient [235] improving anorexia symptomatology.

On the other hand, a disturbed body image promotes severe dietary restriction, weight loss behaviors and also plays a key role in the initiation, persistence and relapse of anorexia [243]. For this reason, some physiotherapy is aimed at self-perception and it must be included in the treatment in combination with other health professionals. For example, mirror exercises, estimation techniques or sensory awareness training have demonstrated their effectiveness as adjunctive intervention to build self-awareness and improved body image distortion [235].

Summarizing, anorexia treatment must be approached from a multidisciplinary point of view where physical therapy and adjunctive therapy may include massage, postural exercises, stretching, breathing and relaxation exercises and other self-perception therapies amongst others. The individualization of the targets of the physical therapy according to the patients characterization and eating disorder severity and course is an essential point.

## 5. Practical Applications

As the main practical application of the present study, we can highlight:-Understanding the individual profile that characterizes each anorexia patient may help in providing a more effective treatment and preventive strategies.-Family support and involvement during the treatment of anorexia patients can have a positive impact on the outcome.-Managing the relationship between the patient and social media will help avoid setbacks in the treatment and avoid any body image influence during and after the treatment.-Eating disorders might be linked to mutations in the transcription factor ESRR α and the transcriptional repressor histone deacetylase 4 (HDAC4). Even so, studies have not proven the relationship between a specific gene and the development of eating disorders.-A diet rich in fiber can improve the diversity of the gut microbiome which is linked to weight gain.-Controlled and supervised physical exercise could help improve the composition of the intestinal microbiota and improve eating disorder symptomatology. It should be included in the patient’s treatment plan.-An analysis of the intestinal microbiota in patients with AN could be useful for nutritional rehabilitation.-Oral care should be included in the treatment of patients with AN, since they have a higher risk of developing gingival and dental disease.-A healthy mouth and smile could help improve the low self-esteem that patients with AN usually present with.-A good clinic history and physical examination are crucial in treating patients with AN.-The rate of weight gain and energy requirements should be calculated individually for each patient.-Nutritional education is a key aspect in treating patients with this disorder.-Each patient needs an individualized nutritional plan.-AN can be accompanied by other psychological disorders, depression being the most common one.-The most used psychological therapy for patients with AN is cognitive behavioral therapy.-Physical therapy can help ease back pain or increase the tone of hypotonic muscles.-Massages can help decrease anxiety and stress, promoting relaxation.

## 6. Conclusions

AN is a psychiatric illness with an unknown etiology where individuals are disturbed by their weight and body shape, without recognizing the graveness of their low weight. The initial consequence of AN is decreased caloric and nutrient intake. With this deficit and malnutrition, disorders appear in different organ systems. Muscles, bones, gut microbiota, as well as the patient’s oral health are seriously affected by this disease. Understanding the profile that characterizes anorexia patients may increase the likelihood of providing more effective treatments and preventive strategies. Family based interventions have proven to be a valuable tool, particularly for children and adolescents, demonstrating promising outcomes in the treatment and recovery of anorexia. Understanding the correlation between social media usage and self-esteem in relation to eating disorders is essential for advancing prevention efforts. Providing a comprehensive clinical history and conducting a thorough physical examination are essential when treating patients with AN. Nutritional education is a fundamental aspect and needs to be a part of all AN treatments. It is recommended to allow the patients to practice physical exercise which can also include massages, as long as this is controlled and supervised.

In conclusion, an individualized treatment for each patient with a multidisciplinary approach is necessary when treating patients with AN. Without the collective work in all of these areas, none of the treatments will be successful.

## Figures and Tables

**Figure 1 nutrients-15-02594-f001:**
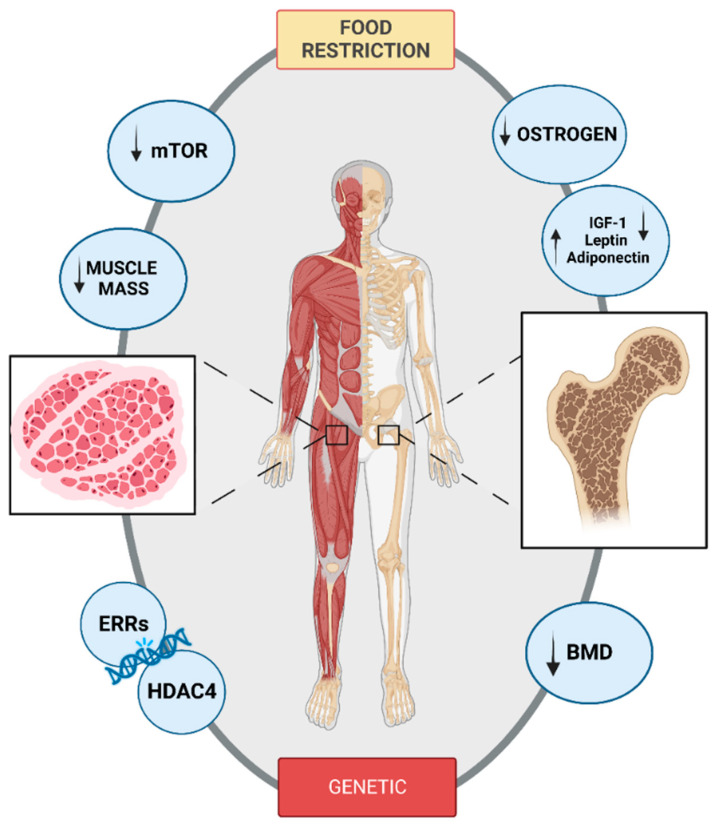
Bone and muscle implications in AN influenced by genetics and food restriction; BMD: bone mineral density; mTOR: mammalian target of rapamycin; HDAC4: histone deacetylase 4; IGF-1: insulin grown factor 1; ERRs: estrogen-related receptors.

**Figure 2 nutrients-15-02594-f002:**
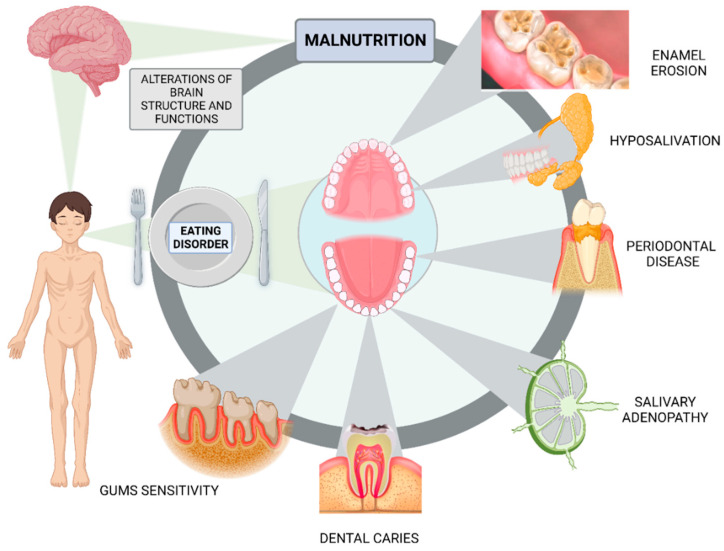
Consequences of malnutrition on the dental health of people suffering from anorexia.

**Table 1 nutrients-15-02594-t001:** Summary of the most relevant benefits according to the intervention.

Author and Year	Study Title	Aim of Study	Main Outcomes and Effectiveness	Duration	Type of Intervention
Marzola et al. (2013) [138]	Nutritional Rehabilitation in Anorexia nervosa: Review of the Literature and Implications for Treatment	To describe issues related to the caloric requirements needed to gain and maintain weight for short and long-term recovery for AN inpatients and outpatients.	The restoration of both nutrient status and weight starts slowly and gradually accelerate as tolerated.	Several weeks	Nutritional
Andrewes et al. (1996) [139]	Computerised psychoeducation for patients with eating disorders	To assess a new computer-based method of health education for patients with bulimia and AN.	The DIET group members were significantly improved when compared to the placebo group in terms of both their knowledge and attitudes towards their disorder.	Not specified	Nutritional
Ng et al. (2013) [140]	Is supervised exercise training safe in patients with anorexia nervosa? A meta-analysis	To examine the effects of supervised exercise training in patients with AN.	Significant improvement in weight and body fat; strength and cardiovascular fitness were also shown to improve.	>2 h/week	Physical Activity
Rizk et al. (2018) [141]	High-intensity exercise is associated with a better nutritional status in anorexia nervosa.	To investigate the links between duration and intensity of exercise and the nutritional status in terms of body composition in acute AN patients.	Exercising at higher intensity in AN is associated with a better nutritional status.	>9 h/week	Physical Activity
Grave et al. (2014) [142]	Inpatient cognitive behavior therapy for adolescents with anorexia nervosa: immediate and longer-term effects.	To establish the immediate and longer-term effects of a novel inpatient program for adolescents that was designed to produce enduring change.	Enhanced cognitive behavior therapy is a promising approach to the treatment of adolescents with severe anorexia nervosa.	20 weeks	Psychological
Steinglass et al. (2012) [143]	Fear of Food as a Treatment Target: Exposure and Response Prevention for Anorexia Nervosa in an Open Series.	To evaluate the potential utility of addressing eating-related fear in the treatment of AN using psychotherapy techniques known to be effective in the treatment of anxiety disorders	Change in anxiety with AN was associated with greater caloric intake	4 weeks	Psychological
Fisher et al. (2018) [144]	Family therapy approaches for anorexia nervosa	To evaluate the efficacy of family therapy approaches compared with standard treatment and other treatments for AN	There was some evidence of a small effect favoring family based therapy compared with other psychosocial interventions in terms of weight gain post-intervention.	No specified	Psychosocial
Barber et al. (2018) [145]	Reducing the Mortality in People with Severe Mental Disorders: The Role of Lifestyle Psychosocial Interventions	To explore the causes of death in high income and low and middle-income countries and review the multi-level risk factor model for mortality in severe mental disorders	Nurse-led services and the utilization of peer support are showing promise outcomes.	>6 months	Psychosocial
Hart et al. (2001) [146]	Anorexia nervosa Symptoms are Reduced by Massage Therapy	To Evaluate massage therapy for women with AN for (1) reducing stress and stress hormone levels, (2) decreasing depression, (3) improving mood, (4) reducing eating disorder symptoms, and (5) increasing dopamine values	Reduced anxiety following their first and last treatment; decreases in body dissatisfaction on the Eating Disorder Inventory and increased dopamine and norepinephrine levels.	5 weeks	Physical Therapy
Fogarty et al. (2013) [147]	Patients with anorexia nervosa receiving acupuncture or acupressure their view of the therapeutic encounter	To investigate the views of patients with AN receiving an acupuncture or acupressure intervention.	Patients perceive the therapeutic relationship and empathy as important qualities of the acupuncture or acupressure intervention as an adjunct therapy for the treatment of AN.	3 weeks	Physical Therapy

## Data Availability

Not applicable.

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
