# Peer review of "The Impact of Anorexia Nervosa and the Basis for Non-Pharmacological Interventions"

_nutrients, 2023, doi:10.3390/nu15112594_

Round 1
Reviewer 1 Report
Thank You for giving me the chance to read Your paper. I found it very interesting and nice to read. My only suggestion is - but it's not a very big deal, as I can say - could You try to change systematic review into a meta-analysis? I believe it could answer three questions: a) how often each type of interventions is being reported? b) how are they effective (or how often are they described as effective)? and c) is there any difference between primary and secondary sources in describing or judging those types of intervention?
The second - minor - suggestion is to shorten or cut these paragraphs describing all biological, psychological, sociological and other correlates of anorexia and its developement. These paragraphs take a large ammount of the paper, but do not align with non-nutrition interventions mentioned in the title. I found these paragraphs well written - comprehensive, clearly systematic, but I believe that reader who wants to read about interventions has some knowledge about etiology etc. If You deside to leave these paragraphs as they are - please consider changes in the title.
Once again thank You for Your valuable work and giving me the chance to read such a good paper.
Reviewer 2 Report
Authors have done an extensive review about anorexia nervosa, their consequences and the main non-pharmacololigical interventions.
Minor modifications must be done:
In all the cases when the name of the author is mentioned in the text, put the number of the reference after the author's name and not at the end of the phrase:
- Page 6, line 284: Levine et al [74]
- Page 6, line 289: Bearman et al [55]
- Page 6, line 299: Rodgers et al [77]
- Page 6, line 301: Levine and Murnen [78]
- Page 6, line 303: Levine and Smolak [73]
- Page 8, line 369: Tiggemann et al [93]
- Page 14, line 652: Roberts and Li [157]
Other issues:
- Page 7, line 350: When authors mentioned: "In this regard, a few years ago, several authors....", please add the corresponding references.
- Page 9, line 363: "....Instagram [92]; among others." Replace the semicolon after [92] by a comma.
- Page 8, line 369: The correct reference is Tiggemann et al [93].
- Page 9, Figure 1. Bone and Muscle implications in AN influenced by genetic and food restriction: All the abbreviations mentioned in the figure should be clarified in the legend independently of the complete name were put in the text.
- Page 10, lines 433, 436, 438 and 439: IGF-1, IGF1, IGF-I refers to the same factor: Insulin-like growth factor 1? If it is the same, please uniform it.
- Page 10, line 454; page 14, lines 650-651 and page 15, line 703: When mentioned "...anorexia nervosa..", replace by AN.
- Page 10, lines 460-461: "Although the bone has received most of the attention at the present, but muscular 460 function is also compromised": Delete the word but before muscular.
- Page 11, line 526: "...around 4.0 1013 bacteria per person in the gut..": I think the amount of bacteria are wrong.
- Page 13, line 586: "...al. [135]and..": Add an space between [135] and "and".
- Page 13, line 603: "...by Kleiman et al. [143,144]: The reference 144 correspond to "Prochazkova et al."
- Page 13, line 615: ".... by Morkl et al. [146,147]: The reference 147 correspond to "Speranza et al.".
- Page 15, line 671: "...(DMFT) ,dental..": Delete the space before the comma and put a new space before "dental".
- Page 18, line 826: "...improve body image concerns [207] even yoga has been proposed..": Delete the word "even" and put a semicolon before "yoga".
- Page 22, line 1062: "..and mindfulness)[206,264,270,271] the physiotherapist should..": Add an space before the numbers of references and a comma before "the physiotherapist"
Reviewer 3 Report
In this article, the authors have reviewed the preventative and non-pharmacological interventions involved in Anorexia nervosa. Multiple hypotheses towards the disease development along with different management strategies have been discussed. With increasing prevalence and complex pathological links makes this review article quite significant. The authors have compiled and reviewed large number of research studies in these fields, and included in this review. This is an interesting review article in my opinion and can be considered for publication. I have listed some of the modifications that should be incorporated prior to final acceptance.
1. Add “nervosa” in your title because your focus is only Anorexia nervosa.
2. Abstract and key words must be improved further.
3. From line 103 to 105, try to avoid upper case word like OR, AND (write these words in lower case)
4. In line 525, 526 please clarify “4.0 1013 bacteria per person in the gut, which is on par with the amount of human cells in the entire body”.
5. Write M. smithii in italics (563, 565) also use correct and full name for first time.
Minor editing of English language required (various spelling and grammatical mistakes were found in this article).
